# Fissure Depth and Caries Incidence in First Permanent Molars: A Five-Year Follow-Up Study in Schoolchildren

**DOI:** 10.3390/ijerph16193550

**Published:** 2019-09-23

**Authors:** Leonor Sánchez-Pérez, María Esther Irigoyen-Camacho, Nelly Molina-Frechero, Marco Zepeda-Zepeda

**Affiliations:** Departamento de Atención a la Salud, Universidad Autónoma Metropolitana (UAM) Unidad Xochimilco, Calzada del Hueso 1100, Ciudad de Mexico 04960, Mexico; meirigo@correo.xoc.uam.mx (M.E.I.-C.); nmolina@correo.xoc.uam.mx (N.M.-F.); mzepeda@correo.xoc.uam.mx (M.Z.-Z.)

**Keywords:** caries risk, fissure morphology, caries increment, fissure depth, follow-up study

## Abstract

This study aimed to evaluate the association between the fissure depth (tooth morphology) of permanent molars and dental caries incidence with a five-year follow-up period. In total, 110 Mexican schoolchildren aged seven years were recruited at baseline, of which 88 children completed the follow-up assessments. The fissure depths of the first permanent lower molars were recorded, and dental caries [decayed, missing, and filled deciduous surfaces (dmfs) and decayed, missing, and filled permanent surfaces (DMFS)] was evaluated annually. A generalized estimating equation model was constructed for evaluating the association between dental caries and fissure depth. The initial and final DMFS scores were 0.02 (±0.2) and 1.55 (±2.49), respectively. The generalized estimating equation model showed that children with deep molar fissures were more likely to develop caries lesions, (Odds Ratio OR = 3.15, *p* = 0.028) compared to children with shallow fissures. Moreover, dental caries in primary teeth (OR = 1.07, *p* = 0.005) was associated with the development of carious lesions in the permanent dentition. Fissure depth was a useful predictor of dental caries, according to this five-year follow-up study. The detection of deep occlusal fissures in the lower first permanent molars could contribute to the identification of children at high risk of dental caries. Tooth morphology may be used to identify children at a higher caries risk, particularly in settings with limited resources.

## 1. Introduction

The incidence of dental caries has declined in some countries. However, it is still highly prevalent [1]. It has been shown that the distribution of dental caries is not homogeneous within a population and a small proportion of children have a high caries index, while the majority of children are “caries-free.” Dental caries is a multifactorial, infectious, and transmissible disease, which is associated with an increase in acid-producing bacteria in the biofilm. Current clinical practice guidelines on caries prevention among schoolchildren indicate sealant and/or fluoride application [1]. 

The use of pit and fissure sealants is difficult in public oral health programs with limited resources, particularly those in low-income countries. In this scenario, the identification of children at higher caries risk is of paramount importance [2,3].

A literature review of risk factors for dental caries in children revealed that more than a hundred factors are significantly associated with dental caries [4,5], which can be grouped into demographic and dietary factors; moreover, factors related to diverse clinical, bacteriological, and physiological parameters have been assessed and applied to different models for caries prediction [5,6,7,8,9,10,11,12]. Past caries experience is considered a strong predictor of caries incidence [5,7,8,9], and the presence of *Streptococcus mutans* and lactobacilli is also associated with increased caries incidence, but bacteriological tests are not very cost-effective. Other possible markers include tooth morphology (pit and fissure depth) [9,10,11].

Evidence suggests that susceptibility to decay is related to the different types of occlusal morphologies [13,14,15]. The first permanent molars show incomplete coalescence of the fissures, permitting retention of the biofilm at the base of the defect, which increases the risk of the development of caries lesions. It is also known that both occlusal morphology and enamel defects follow genetic and familial patterns [16], modulated by hygiene and sugar intake.

For almost a decade, it has been suggested that the incidence of dental caries has increased in deciduous teeth in several countries, thereby increasing the risk of the development of tooth decay in the permanent teeth [17]. This supports the need for newer, simpler, and more accurate risk indicators to identify individuals at higher risk of developing caries. In 2011, Leroy et al. suggested that longitudinal studies are essential to identify risk factors, as by definition, a risk factor must clearly establish that the exposure has occurred before the outcome [5]. 

To contribute to the collection of evidence for future clinical guidelines, we tested the hypothesis that deep pits and fissures in the first permanent molars are associated with caries incidence among schoolchildren, who were followed for five years. 

## 2. Materials and Methods

### 2.1. Participants

In 2009, a cohort study of seven-year-old children from a public (state-funded) school in southern Mexico City was initiated. The school was situated in a medium-income area. A total of 135 children were registered at the surveyed school. Before taking any measurements, written informed consent was obtained from the parents, and 110 children (53 boys and 57 girls) participated in the study. No data were collected from the 25 children whose parents failed to provide consent for participation. After a five-year follow-up period, 88 children (44 boys and 44 girls) remained in the study, with a 20% dropout rate. Figure 1 depicts the yearly dropout rate. Shifting residence or changes in the parent’s workplace were the main reasons for changing schools or dropping out of the cohort. No statistically significant difference was observed in the baseline dental caries index between the children who remained in the cohort (*n* = 88) and those who dropped out (*n* = 22), (*p* > 0.05). Only children with complete data for each annual dental evaluation were included in this study. 

Ethical Considerations

The study protocol was reviewed and approved by the Autonomous Metropolitan University-Xochimilco (IRB: 15/93, 14/10, 10/17, CE.002.17). All the children received oral health-education sessions every year and toothbrushes were distributed free of charge every six months. 

### 2.2. Fissure Depth

The fissure depths of both lower first permanent molars were recorded at baseline [13,15]. A World Health Organization (WHO) periodontal probe was used to assess fissure depth [18]. An examination head light (Lumindex™ 5 LED, Dentmate Technology Co., LTD, Taipei, Taiwan) was used to illuminate the oral cavity during examination. Morphologically, the occlusal surface of molars and premolars is characterized by the presence of grooves that correspond to the zone of confluence of the developmental lobes. Fissures can be observed at the junction of the cusps of posterior teeth. According to Köning, the cusp inclines in shallow fissures (SF), forming a “V-angle” greater than 70° [14], and the tip of a WHO-type periodontal probe can reach the base of the fissure. The bottom of the fissure can be observed in good light. Deep fissures (DF) appear slit-like, the cuspal inclines form a “V-angle” lower than 70°, and the tip of the probe cannot penetrate the fissure. The base of this type of fissure is not visible when examined in good light. Figure 2 illustrates the “V-angle” in posterior permanent molars. The same researcher (LS-P) classified the fissures after probing them, to avoid bias. 

### 2.3. Calibration

The calibration exercise consisted of two phases. The theoretical phase involved a discussion of the diagnostic criteria for fissure depth, as well as photographic analysis. The clinical phase was performed with randomly selected public schoolchildren, who had not participated in the study. The researcher (LS-P) examined 25 selected children; intra-examiner agreement was tested by re-examining and comparing the observations after a 14-day interval. The Kappa coefficient was 0.90 and the examiner was considered competent to conduct the study. 

### 2.4. Caries and Oral Hygiene

Dental caries indices [decayed, missing, and filled deciduous surfaces (dmfs) and decayed, missing, and filled permanent surfaces (DMFS)] were calculated as recommended by the World Health Organization (WHO) [18]. Radiographs were not obtained. 

To determine the caries increment, the same examiners repeated the dental examination annually. To prevent observer bias, the children were evaluated without access to their caries records. For each child, the caries increment was calculated with the baseline dmfs or DMFS score and the last available dmfs or DMFS score, without considering reversals. Oral hygiene was evaluated using the Simplified Oral Hygiene Index (OHI-S) [19].

### 2.5. Statistical Analysis

Bivariate analysis was performed using the chi-squared and student’s t-tests to determine the relationship between fissure depth and caries prevalence in permanent teeth (DMFS > 0) during the follow-up. For comparison of categorical variables, the Chi-squared test or Fisher-exact test was applied. Generalized estimating equation (GEE) models were constructed to adjust for the clustering effect of longitudinal observations, using the dental caries index score (dichotomized as DMFS = 0 and DMFS > 0) as the dependent variable and fissure depth, dental caries in primary teeth, and oral hygiene as independent variables, after controlling for age and sex. For model construction, logit and binary were used as the link function and the family, respectively. Odds ratios (OR) and 95% confidence intervals (CI) were obtained from the models. The interactions were tested within the models. The test quasi-likelihood under the independence model criterion (QIC) was used to identify the best GEE model for the set of covariates studied. While comparing the models, lower QIC figures indicated a better fit [20]. Moreover, the Wald test was used to evaluate the significance of the covariates included in the models. A *p*-value < 0.05 was considered statistically significant. The STATA v12 statistical package (College Station, TX, USA) was used for data analysis. 

## 3. Results

A total of 110 children were included in the initial oral examination. At the end of the five-year follow-up, 88 children (50% girls) remained in the cohort. Caries prevalence was 60.2% (dmfs + DMFS > 0) at the first examination. No significant differences in the caries index scores were detected between the sexes (*p* = 0.276). The mean OHI-S score was 1.5 (±0.39). Approximately half of the children (51.2%) had OHI-S scores ≥ 1.50 and were considered to have poor oral hygiene, and 48.9% had a lower biofilm accumulation (OHI-S < 1.50) and comprised the good oral hygiene group. 

At the beginning of the study, the dmfs indices were high in both boys [dmfs 5.4 (±8.5)] and girls [5.1 (±5.8)] (*p* = 0.861). The mean dmfs for each year of observation was 5.23 (±7.26), 5.44 (±7.56), 5.08 (±6.57), 3.88 (±6.0), 2.52 (±4.86), and 1.38 (±3.60), from the first to the last year. At the beginning of the study, the DMFS was 0 and 0.05 (±0.2) for boys and girls, respectively (*p* > 0.050). At the end of the follow-up, the mean DMFS was 1.45 (±2.32) and 1.64 (±2.47) for boys and girls (*p* = 0.734), respectively (Table 1). Oral hygiene was associated with dental caries in primary teeth, at baseline. Children with poor oral hygiene had a mean dmfs score of 2.51 (±3.43), and a mean dmfs score of 7.82 (±8.88) was observed in children with good oral hygiene (*p* < 0.001). 

The mean DMFS score for each year of observation was 0.02 (±0.15), 0.15 (±0.36), 0.30 (±0.96), 0.59 (±1.53), 0.82 (±1.80), and 1.55 (±2.49), from the first to the last year. Children with caries-free permanent teeth had good oral hygiene (OHI-S < 1.50), and the two children with dental caries in the permeant teeth at baseline had poor oral hygiene (OHI-S ≥ 1.50). Table 2 shows the percentage of first permanent molars that presented with new carious lesions during the follow-up period. The lower right first molar was the tooth most frequently affected by caries: Close to one-third of these teeth developed new lesions (31.8%) and the least affected was the upper right permanent molar (12.5%). 

It was found that 86.4% of the children had shallow fissures and 13.6% had deep fissures, without significant sex-based differences (*p* = 0.801). Table 3 presents the caries prevalence in the permanent dentition at the end of the follow-up based on fissure depth. A higher percentage of carious lesions was found in the group with deep fissures (75%), compared to the group with shallow fissures (39.5%), (*p* = 0.023). 

Figure 3 depicts the DMFS index by fissure depth during the follow-up. The yearly caries increment for the cohort, according to the DMFS index, was 0.13, 0.16, 0.28, 0.23, and 0.75, from the first through the last year of follow-up. Therefore, the highest increment was found between the fourth and the fifth year of observation, during the transition between the ages of 11 and 12.

Only the children with deep fissures [DMFS = 0.20 (±0.2)] in the lower permanent first molars had caries at the beginning of the study. Statistically significant differences were observed for DMFS scores when comparing children with shallow and deep fissures for each year of the study; at the end of the study, children with deep fissures had higher DMFS scores (3.17) than children with shallow fissures (1.30) (*p* = 0.004). 

Moreover, higher caries index scores were observed at baseline in the primary teeth, based on fissure depth; children with shallow fissures had a dmfs score of 3.2 (±5.9) and those with deep fissures had scores of 9.8 (±9.6) (*p* = 0.013). A higher prevalence of caries in the permanent teeth was detected in children with deep fissures. 

Table 4 presents the results of the multivariate GEE models. It was observed that sex, age, and oral hygiene at the beginning of the study were not significantly associated with the development of dental caries in permanent teeth during the five-year follow-up. In the three models, the dmfs index was associated with caries in the permanent teeth, in the first, second, and third models (OR = 1.07, *p* = 0.014; OR = 1.07, *p* = 0.005; OR = 1.09, *p* < 0.001, respectively). The QIC statistic showed that model 2 had the best fit; dental caries in the primary teeth and fissure depth, age, and sex were included. The results indicated that children with deep fissures were 3.15 times (*p* = 0.028) as likely to develop dental caries in the permanent teeth compared to children with shallow fissures. The QIC of GEE model 3 was the highest of the constructed models, indicating that its goodness of fit was poorer than those of the other two fitted models. Fissure depth was excluded from this model (Table 4).

## 4. Discussion

In this five-year follow-up study, fissure depth in the lower first permanent molars was associated with caries incidence; children showing deep fissures were three times as likely to develop dental caries (OR 3.15) compared to children with shallow fissures, after adjusting for dmfs, age, and sex at baseline. In schoolchildren, the occlusal fissures of the first permanent molars are the most susceptible to dental caries [9,10]. Moreover, it has been suggested that caries susceptibility depends on the morphology of this surface [2,11,14,21].

In the present study, the lower first permanent molars showed a higher caries incidence than their upper counterparts, where more than a quarter of the lower molars developed caries. Similar results have been reported in other groups of children [22,23,24,25]. The first permanent molars have a long eruption time, and dentists must carefully examine these teeth during this period and make the patient aware of the caries risk to their newly erupted molars [11,24,26]. 

The higher susceptibility to dental caries in the group of children with deep fissures is consistent with the complex anatomy of fissures found in the grooves systems. A study using microradiographs of fissures, enamel grooves, and their contents concluded that the internal morphology of the interlobar grooves influences the conditions for bacterial growth, which determines the location of caries progression within the fissure–groove system. Low caries activity at the deepest portion of the grooves implies a low level of bacterial viability at these sites [27]. Moreover, as described by Loesche and Sreaffon [28], the number of bacteria on the occlusal fissures is a suitable indicator of the number of cariogenic bacteria in the mouth. It has been shown that caries occurs in the area surrounding the fissure entrance, rather than the base of the fissure [2,21].

Most of the children in this study (86%) had shallow fissures and only 14% had deep fissures. This may be the case in other groups of children [29]. The application of pit and fissure sealants would be a good option for children with deep fissures, at medium or high risk for caries [8].

Dental caries is a multifactorial disease, and different risk factors should be considered for determining the best treatment for diverse groups of patients. The GEE results showed that the best prediction model was obtained when both the caries experience in primary teeth and fissure depth in first permanent molars were included. Dental caries in primary teeth has been considered among the best predictors of dental caries in permanent teeth, because multiple factors associated with dental caries, such as a high-carbohydrate diet, poor oral hygiene, and the biofilm, interact to produce caries in the primary and permanent teeth in the child’s mouth. Moreover, the results showed that fissure depth contributes to the prediction of caries, despite the presence of caries in the primary teeth included in the model. This may have clinical implications, as dentists could consider both factors while formulating the best dental treatment plan for their patients. Moreover, the implementation of public dental caries-preventive programs may be beneficial when both aspects are considered to identify high-risk groups. The assessment of fissure depth and caries experience in primary teeth is cost-effective and less time-consuming. This contrasts with costly laboratory techniques used to identify caries risk. The application of simple risk-markers would improve the cost-effectiveness of the caries preventive programs, which is particularly important in low-income communities.

In this study, caries incidence was higher in children aged 11 to 12 years than younger children. Similar findings were reported in other follow-up studies [11,30]. The high-caries increment in this age group may be associated with the changes experienced during this period of life, including pre-adolescence, considering that children enter a rapid growth period related to adolescence; they face important changes associated with variations in hormonal levels and changes in eating habits and hygiene practices. However, other studies have found high caries increments in children aged seven to nine years. A three-year follow-up study with 10- to 13-year-old children found caries increments of 20%, 5%, and 7% in the first, second, and third years of observation, respectively. It is possible that the period of highest caries incidence in schoolchildren varies according to eating habits, oral hygiene practices, and access to dental services, among other factors [31].

One of the limitations of the study was its small sample size; however, the patients were followed-up for five years and the attrition rate was low. The participants came from a medium-socioeconomic-status county and had eating habits and access to dental services similar to many children in Mexico City; however, caution is needed while generalizing the results of the present study to other groups of children with different ethnic backgrounds, which could influence fissure morphology, among other factors. Despite the small sample size, it was possible to detect a large difference in caries incidence between children with deep fissures compared to those with shallow fissures. 

## 5. Conclusions

The children studied had a high dental caries index in their primary teeth, and this index was associated with caries incidence in the permanent teeth after five years of follow-up. Moreover, the newly developed caries lesions in the permanent dentition were associated with fissure depth in the lower first permanent molars. Public health programs could incorporate the evaluation of fissure morphology to identify children at high-caries risk. Dentists should be aware of the importance of the fissure anatomy and select the most appropriate preventive method for these patients. 

## Figures and Tables

**Figure 1 ijerph-16-03550-f001:**
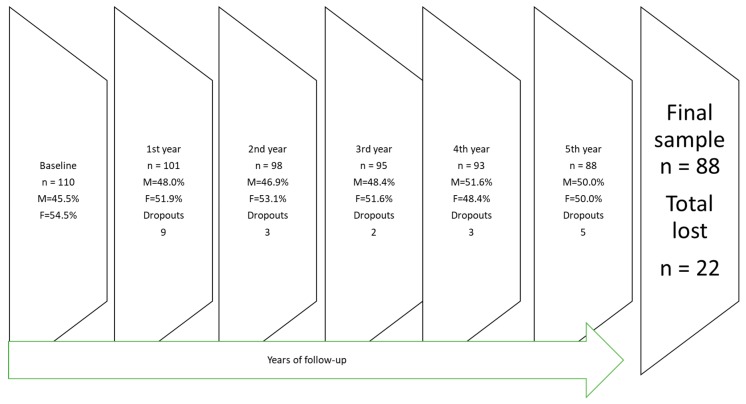
Number of children lost during follow-up during each year of the study.

**Figure 2 ijerph-16-03550-f002:**
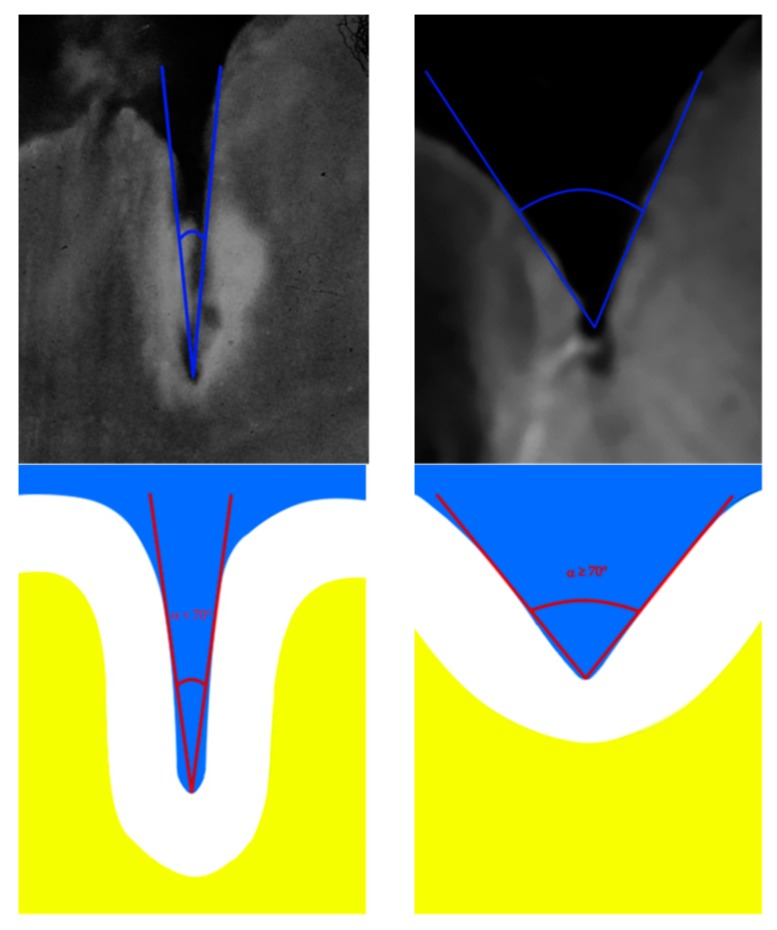
Shallow and deep fissures. Less than 70° “V angle” indicates deep fissure and a wider than 70° “V angle” a shallow fissure. [14].

**Figure 3 ijerph-16-03550-f003:**
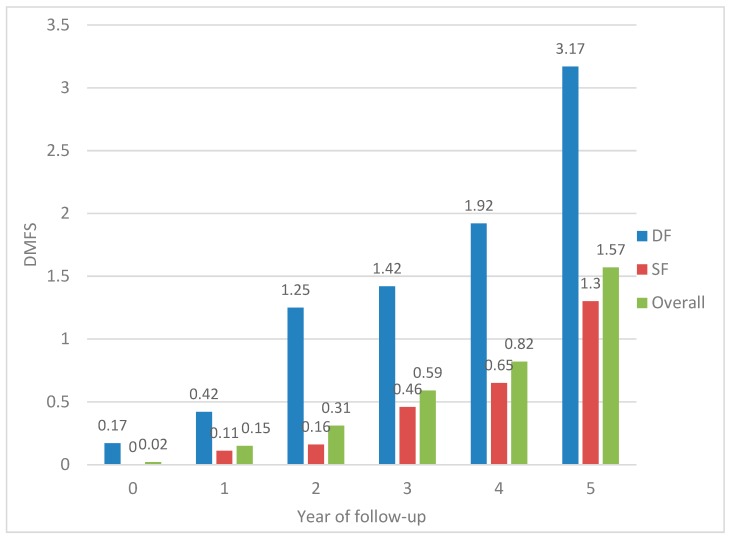
Baseline and annual mean DMFS index: Overall and based on fissure depth. DMFS = decayed, missing, and filled permanent surfaces; DF = deep fissure; SF = shallow fissure.

**Table 1 ijerph-16-03550-t001:** Distribution of clinical markers in the follow-up, based on sex.

Year of Study	Present Teeth		Caries Indexes	
	Deciduous	Permanent	dmfs	DMFS
	Mean (SD)	Mean (SD)	Mean (SD)	Mean (SD)
Baseline				
Boys	16.1 (2.5)	6.7 (3.4)	5.4 (8.5)	-
Girls	15.5 (2.4)	7.4 (2.8)	5.1 (5.8)	0.05 (0.2)
1st year				
Boys	13.6 (2.5)	9.1 (3.0)	5.8 (9.0)	0.07 (0.3) *
Girls	13.1 (2.5)	10.1 (2.5)	5.1 (5.9)	0.2 (0.4)
2nd year				
Boys	12.1 (2.2)	11.4 (2.5)	5.4 (7.7)	0.2 (0.5)
Girls	11.5 (2.8)	12.2 (2.8)	4.7 (5.3)	0.4 (1.3)
3rd year				
Boys	9.3 (3.7)	14.2 (4.1)	4.6 (7.0)	0.4 (1.0)
Girls	8.3 (4.4)	15.6 (4.5)	3.2 (4.8)	0.8 (1.9)
4th year				
Boys	6.2 (4.0)	17.7 (4.8)	2.7 (5.6)	0.7 (1.3)
Girls	5.5 (4.8)	18.7 (5.4)	2.3 (4.1)	1.0 (2.2)
5th year				
Boys	2.3 (3.2)	23.1 (4.6)	1.5 (4.1)	1.5 (2.7)
Girls	2.5 (3.0)	22.8 (4.4)	1.3 (3.1)	1.6 (2.3)

dmfs = decayed, missing, and filled deciduous surfaces, DMFS = decayed, missing, and filled permanent surfaces * = Statistically significant differences with student’s *t*-test.

**Table 2 ijerph-16-03550-t002:** New caries lesions in first permanent molars during each year of follow-up.

1st Molar (*n* = 88)	Year of Examination							
	Baseline	1st	2nd	3rd	4th	5th	Total	%
		Caries Lesions						
Upper right	0	0	1	3	1	6	11	12.5
Upper left	1	0	2	2	2	9	16	18.2
Lower left	2	2	4	4	2	8	22	25.0
Lower right	0	3	5	6	3	11	28	31.8
Total	3	5	12	15	8	34	77	21.9

**Table 3 ijerph-16-03550-t003:** Baseline fissure depth in the lower first permanent molars and dental caries prevalence during the final observation.

Fissure Depth	DMFS > 0		DMFS ≥ 1		*p* *	Total	
	*n*	(%)	*n*	(%)		*n*	(%)
Shallow	46	(60.5%)	30	(39.5%)	0.023	76	(86.4%)
Deep	3	(25.0%)	9	(75.0%)		12	(13.6%)

* Fisher’s exact test.

**Table 4 ijerph-16-03550-t004:** Results of the bivariate and multivariate generalized estimating equation (GEE) models using baseline information of fissure depth, oral hygiene, and dental caries in primary teeth as predictor variables and the outcome variable incidence of dental caries (DMFS > 0), in schoolchildren followed-up for five years.

**Characteristic**	**Crude OR**	**(95% CI)**	***p***
Age	1.31	(0.46, 3.72)	0.648
Sex (female) ^a^	1.36	(0.75, 2.46)	0.310
S-OHI (>1.5) ^b^	1.83	(0.92, 3.63)	0.085
dmfs index	1.05	(1.02, 1.07)	0.001
Fissure depth (deep) ^c^	2.73	(1.59, 4.72)	0.001
**Model 1 ^d^**	**Adjusted OR**	**(95% CI)**	***p***
Age	1.25	(0.44, 3.58)	0.678
Sex (female) ^a^	1.60	(0.80, 3.19)	0.184
S-OHI (>1.5) ^b^	1.08	(0.48, 4.41)	0.851
dmfs index	1.07	(1.01, 1.13)	0.014
Fissure depth (deep) ^c^	3.11	(1.14, 8.47)	0.026
**Model 2 ^e^**	**Adjusted OR**	**(95% CI)**	***p***
Age	1.25	(0.44, 3.62)	0.670
Sex (female) ^a^	1.59	(0.81, 3.15)	0.184
dmfs index	1.07	(1.02, 1.13)	0.005
Fissure depth (deep) ^c^	3.15	(1.13, 8.78)	0.028
**Model 3 ^f^**	**Adjusted OR**	**(95% CI)**	***p***
Age	1.36	(0.43, 4.30)	0.598
Sex (female) ^a^	1.70	(0.87, 3.35)	0.123
dmfs index	1.09	(1.04, 1.14)	0.001

Reference categories: ^a^ sex = male, ^b^ Simplified Oral Hygiene Index S-OHI ≤ 1.5, ^c^ fissure depth = shallow, **^d^** Model 1 quasi-likelihood under the independence model criterion (QIC) = 513.9 (Wald test = 20.9, *p* < 0.001), **^e^** Model 2, QIC = 508.6 (Wald test = 20.6, *p* < 0.001), **^f^** Model QIC = 557.5 (Wald test = 2.96, *p* = 0.085).

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
