# Peer review of "Fissure Depth and Caries Incidence in First Permanent Molars: A Five-Year Follow-Up Study in Schoolchildren"

_ijerph, 2019, doi:10.3390/ijerph16193550_

Round 1
Reviewer 1 Report
The authors present the results of their original research focussing on the incidence of caries in schoolchildren. They assess easy-to-obtain fissure depth as a potential risk factor.
The topic is of special interest. After moderate English changes and minor revision, I recommend to accept this manuscript.
Please use always the same number of decimals (e.g. three) when reporting p-values
Abstract:
- Language "The aim of this study was evaluate the association between fissure depth in permanent molars and dental caries incidence in a five year-follow up study."
- Briefly, how do you define "depth molar fissures"?
Introduction:
- Language "Predictive models can help direct the limited resources available to those children in greater need."
Materials and methods:
- "Changing the school of attendance was the main reason for dropping out from the cohort" please report all reasons for dropout including corresponding number of children
- Please state IRB approval number
- "the cusp inclines formed an angle narrower the base of the fissure appeared to be visible under illumination" Which source of illumination was used?
- Maybe a drawing could demonstrate the difference between both shallow and depth fissures.
- Language: "The Kappa coefficient was 0.90 and the examiner were considered able to perform the study." Just one examiner, therefore it must read "was considered"
- Please cite reference for "The caries index (dmfs and DMFS) was calculated as recommended by the World Health Organization (WHO)." How were multiple observations at the same surface (e.g. secondary caries) scored?
Results:
- Please state overall fit of the multivariate GEE models
Figure 1:
- Language: Please write "1st" instead of "1rst" and "4th" instead of "4rd"
- Writing error: "5405%". Maybe 54.5%?
- Writing error: It cannot be n=98 in the 5th year because you have had only n=93 in the 4th year. Maybe n=88?
Figure 2:
- Do NOT draw lines between data-points. You have absolutely no information regarding the time between your measurements.
Table 1:
- Language: Please write "2ns" instead of "2scnd" and "3rd" instead of "3rth"
- Please start counting with "baseline" followed by "1st year" (the same way as shown in Figure 1)
Table 3:
- Please write "DMFS>0" instead of "DMFS≥1" (as written in the text)
- Please write % in ()
Table 2:
- Please use same spelling for "baseline"
- Language: Write "4th" instead of "4rd"
Author Response
All changes, insertions or modifications are written in red in the text.
REVIEWER 1
-Please use always the same number of decimals (e.g. three) when reporting p-values
We have included three places after the decimal, while reporting all the p-values. (Pages 5,6,7 line: 151, 155, 157, 159, 161, 177, 180, 198, 200).
Abstract:
- Language "The aim of this study was evaluate the association between fissure depth in permanent molars and dental caries incidence in a five year-follow up study."
We have corrected the language of this sentence:
This study aimed to evaluate the association between fissure depth (tooth morphology) of the first permanent molars and the dental caries incidence with a five-year follow-up study" (Page 1, line 18-19).
Your question:
- Briefly, how do you define "depth molar fissures"?
Morphologically, the occlusal surface of molars and premolars, are characterized by the presence of grooves that correspond to the zone of confluence of the developmental lobes. Fissures can be observed at the junction of the cusps of posterior teeth. According to Köning, the cusp inclines in shallow fissures (SF), forming a “V-angle” greater than 70°, [14] and the tip of a WHO-type periodontal probe can reach the base of the fissure. The bottom of the fissure can be observed in good light(an examination head light Lumindex™ 5 LED, Dentmate Technology Co., LTD, Taipei, Taiwan). Deep fissures (DF) appear slit-like, the cuspal inclines form a “V-angle” lesser than 70°, and the tip of the probe cannot penetrate the fissure. The base of this type of fissure is not visible when examined in good light. Figure 1 illustrates the “V angle” in posterior permanent molars. Figure 2 illustrates the “V-angle” in posterior permanent molars. (Figure 2, page 4 ).
Introduction
- Language "Predictive models can help direct the limited resources available to those children in greater need."
"Predictive models can help to direct the available resources to children in greater need." (page 1, line: 68-69)
Materials and methods:
-"Changing the school of attendance was the main reason for dropping out from the cohort" please report all reasons for dropout including corresponding number of children
The reasons for dropping out were shifting residence, divorce, and changes in the parent’s work; however, the school does not provide this information. Thus, we cannot provide the exact reasons for dropping-out. However, we performed a statistical analysis and there were no significant differences in the baseline sociodemographic variables and dental caries score of the children who completed the study and those who dropped out.
-Please state IRB approval number Institutional Review Board (IRB)
The University IRB protocol approval number has been provided in the revised manuscript. The university has a review board, which reviews and approves the ethical aspects of the research protocol and determines if the study fulfils the corresponding requirements and complies with legislation. (Page 3, line 92)
- "the cusp inclines formed an angle narrower the base of the fissure appeared to be visible under illumination" Which source of illumination was used?
We used an examination head light (Lumindex™ 5 LED, Dentmate Technology Co., LTD, Taipei, Taiwan) as the source of illumination. We have added this information in the revised manuscript (page 2, line 98)
- Maybe a drawing could demonstrate the difference between both shallow and depth fissures.
We think that this was an excellent suggestion. An illustration was added (Figure 3) in the revised manuscript (page 4)
-Language: "The Kappa coefficient was 0.90 and the examiner were considered able to perform the study." Just one examiner, therefore it must read "was considered"
"The Kappa coefficient was 0.90 and the examiner was considered competent to conduct the study."(page 4, line 122)
- Please cite reference for "The caries index (dmfs and DMFS) was calculated as recommended by the World Health Organization (WHO)."
The WHO reference was included (Page 4 line:126) as follows:
World Health Organization. Oral Health Surveys-basic Methods. 4th ed. Geneva: World Health Organization; 1997.
-How were multiple observations at the same surface (e.g. secondary caries) scored?
We followed the WHO criteria for each oral examination. If a secondary caries was observed on a filled tooth surface, it received a score of 2 (filled, with decay).
Results:
- Please state overall fit of the multivariate GEE models
At present, there is no accepted test for determining the overall fit for GEE models, using binomial distributed outcome variables. However, in recent years, the ‘quasi-likelihood under the independence model criterion (QIC)’ was developed. This criterion allows the selection of the best subset of covariates in the models[1]. We have added this information to the methods section in the revised manuscript (page 4 line 139). We used this criterion and reported the results in Table 4 of the revised manuscript. Moreover, we included the results of the Wald test in Table 4, which provides information on the significance of the predictors in the model.
- Language: Please write "1st" instead of "1rst" and "4th" instead of "4rd"
- Writing error: "5405%". Maybe 54.5%?
- Writing error: It cannot be n=98 in the 5th year because you have had only n=93 in the 4th year. Maybe n=88?
We have corrected the abbreviations for first and fourth as 1st and 4th, respectively and changed n=98 to n=88 for the 5th year of study, in Figure 1 . We have corrected the erroneous figure 5405% to 50.1%.
Figure 2:
- Do NOT draw lines between data-points. You have absolutely no information regarding the time between your measurements.
We change the type of graph in Figure 3, as per the reviewer’s suggestion. We prepared a bar chart, which has been added to the revised version of the manuscript (page 7).
Table 1:
- Language: Please write "2ns" instead of "2scnd" and "3rd" instead of "3rth"
- Please start counting with "baseline" followed by "1st year" (the same way as shown in Figure 1)
We have used 2nd and 3rd, as suggested and included the information presented in Figure 1 in the first column.
Table 2:
- Please use same spelling for "baseline"
- Language: Write "4th" instead of "4rd"
We have corrected the spelling for the word ‘baseline’, and corrected 4rd to 4th.
Table 3:
- Please write "DMFS>0" instead of "DMFS≥1" (as written in the text)
- Please write % in ()
We have used "DMFS>0" instead of "DMFS≥1" (as written in the text).
[1] Cui, J. QIC program and model selection in GEE analysis. Stata J. 2007, 7,209-220.

Reviewer 2 Report
Caries is a multifactorial disease, related to tooth morphology, dietary, brushing habits, quality of saliva, etc. So, if you evaluate only the tooth morphology, I am afraid there is a bias. I was wondering if the authors have evaluated and isolated the other parameters that are related to caries risk since it is not clarified in the text.
Also, I would like to suggest the following articles to be included in the reference list of the paper.
X.-L. Gao, C.-Y.S. Hsu, Y. Xu, H.B. Hwarng, T. Loh and D. Koh, Building Caries Risk Assessment Models for Children, Journal of Dental Research, 10.1177/0022034510364489, 89, 6, (637-643), (2010).
Roos Leroy, Kris Bogaerts, Luc Martens and Dominique Declerck, Risk factors for caries incidence in a cohort of Flemish preschool children, Clinical Oral Investigations, 16, 3, (805), (2012).
Author Response
ANSWERS TO REVIEWERS’ COMMENTS (IJERPH_584672)
Annotations:
Cursive = Reviewer
Bold = Answers
All changes, insertions or modifications are written in red in the text.
REVIEWER 2
Comments and Suggestions for Authors
-Caries is a multifactorial disease, related to tooth morphology, dietary, brushing habits, quality of saliva, etc. So, if you evaluate only the tooth morphology, I am afraid there is a bias. I was wondering if the authors have evaluated and isolated the other parameters that are related to caries risk since it is not clarified in the text.
We have included the analysis of oral hygiene with the Simplified Oral Hygiene Index (OHI-S) in the revised manuscript. This variable was not statistically significant in the GEE model and was thus, omitted in the previous version of the manuscript; however, we agree that the origin of dental caries is multifactorial and oral hygiene is an important factor for caries development.
-Also, I would like to suggest the following articles to be included in the reference list of the paper.
X.-L. Gao, C.-Y.S. Hsu, Y. Xu, H.B. Hwarng, T. Loh and D. Koh, Building Caries Risk Assessment Models for Children, Journal of Dental Research, 10.1177/0022034510364489, 89, 6, (637-643), (2010).
Roos Leroy, Kris Bogaerts, Luc Martens and Dominique Declerck, Risk factors for caries incidence in a cohort of Flemish preschool children, Clinical Oral Investigations, 16, 3, (805), (2012).
These references have enriched the manuscript and have been included in the introduction to the revised version of this manuscript.

Reviewer 3 Report
Need for English grammar and style revisions.

Author Response
ANSWERS TO REVIEWERS’ COMMENTS (IJERPH_584672)
Annotations:
Cursive = Reviewer
Bold = Answers
All changes, insertions or modifications are written in red in the text.
REVIEWER 3
-Suggestion for English grammar and style revisions sent to Editor
This reviewer indicated that several sections of manuscript could be improved. We have revised the entire manuscript and made the necessary changes suggested by Reviewers 1 and 2. We have also enrolled the help of an English-language editing service for editing the revised manuscript.

Round 2
Reviewer 3 Report
Paper is much improved. I made a few grammar suggestions to the revised draft.
